# Toward a Physiologically Relevant 3D Helicoidal-Oriented Cardiac Model: Simultaneous Application of Mechanical Stimulation and Surface Topography

**DOI:** 10.3390/bioengineering10020266

**Published:** 2023-02-17

**Authors:** Fatemeh Navaee, Philippe Renaud, Niccolò Piacentini, Mathilde Durand, Dara Zaman Bayat, Diane Ledroit, Sarah Heub, Stephanie Boder-Pasche, Alexander Kleger, Thomas Braschler, Gilles Weder

**Affiliations:** 1Microsystems Laboratory-LMIS4, EPFL, 1015 Lausanne, Switzerland; 2Department of Pathology and Immunology, Faculty of Medicine, CMU, 1206 Geneva, Switzerland; 3Institute of Molecular Oncology and Stem Cell Biology, Ulm University Hospital, 89081 Ulm, Germany; 4Swiss Center for Electronics and Microtechnology (CSEM), 2002 Neuchatel, Switzerland; 5Interdisciplinary Pancreatology, Department of Internal Medicine 1, Ulm University Hospital, 89081 Ulm, Germany; 6Organoid Core Facility, Medical Faculty, Ulm University Hospital, 89081 Ulm, Germany

**Keywords:** cardiac model, helicoidal orientation, 3D cell orientation, mechanical stimulation, surface topography

## Abstract

Myocardium consists of cardiac cells that interact with their environment through physical, biochemical, and electrical stimulations. The physiology, function, and metabolism of cardiac tissue are affected by this dynamic structure. Within the myocardium, cardiomyocytes’ orientations are parallel, creating a dominant orientation. Additionally, local alignments of fibers, along with a helical organization, become evident at the macroscopic level. For the successful development of a reliable in vitro cardiac model, evaluation of cardiac cells’ behavior in a dynamic microenvironment, as well as their spatial architecture, is mandatory. In this study, we hypothesize that complex interactions between long-term contraction boundary conditions and cyclic mechanical stimulation may provide a physiological mechanism to generate off-axis alignments in the preferred mechanical stretch direction. This off-axis alignment can be engineered in vitro and, most importantly, mirrors the helical arrangements observed in vivo. For this purpose, uniaxial mechanical stretching of dECM-fibrin hydrogels was performed on pre-aligned 3D cultures of cardiac cells. In view of the potential development of helical structures similar to those in native hearts, the possibility of generating oblique alignments ranging between 0° and 90° was explored. Indeed, our investigations of cell alignment in 3D, employing both mechanical stimulation and groove constraint, provide a reliable mechanism for the generation of helicoidal structures in the myocardium. By combining cyclic stretch and geometric alignment in grooves, an intermediate angle toward favored direction can be achieved experimentally: while cyclic stretch produces a perpendicular orientation, geometric alignment is associated with a parallel one. In our 2D and 3D culture conditions, nonlinear cellular addition of the strains and strain avoidance concept reliably predicted the preferred cellular alignment. The 3D dECM-fibrin model system in this study shows that cyclical stretching supports cell survival and development. Using mechanical stimulation of pre-aligned heart cells, maturation markers are augmented in neonatal cardiomyocytes, while the beating culture period is prolonged, indicating an improved model function. We propose a simplified theoretical model based on numerical simulation and nonlinear strain avoidance by cells to explain oblique alignment angles. Thus, this work lays a possible rational basis for understanding and engineering oblique cellular alignments, such as the helicoidal layout of the heart, using approaches that simultaneously enhance maturation and function.

## 1. Introduction

Native human myocardium features cardiac cells within a dynamic environment that respond to physical, biochemical, and electrical stimulations. This dynamic architecture is vital for cardiac tissue metabolism, function, and physiology. Parallel orientations of cardiomyocytes within the myocardium result in a local dominant myocyte orientation [1,2]. The local alignment of fibers, along with the main stress and strain direction, are substantiated by a complex helicoidal organization that becomes evident at the macroscopic scale (Figure 1). Indeed, the heart muscle twists along its long axis during ejection, an event that is supported by helical rotation in the fiber alignment of tissue (Figure 1) [1,3,4,5,6,7]. Altogether, this translates in a wringing motion which helps ejection. The twist motion is critical for heart function as, without this torsion, around 20% of myocardial contraction during each cycle would, at best, translate to 20% circumferential contraction and a corresponding ejection fraction ranging between 15 and 20%. Given the particular helicoidal arrangement, the myocardial contraction performs at 60–70% capacity in physiologic conditions [8]. Quantitatively, the rotation degree is major. While the subendocardial fibers are dispersed in a 60° angle along the long axis, the subepicardial fibers run at an angle of 60° in the opposite direction [8]. Any cardiac insufficiency leading to the alteration of heart shape substantially impacts not only the rotation of heart, but also the angles of the myocardial fibers [8], ultimately leading to remodeling, reorganization, heart failure, or even death [1]. During cellular relaxation in diastole, cardiomyocytes become passively elongated, while they actively resolve their high load during systolic contractions. At the same time, a certain static stress is always present due to the incomplete depletion of ventricles. Mathematic models of cardiac mechanics, corroborated with experimental studies on animal cardiac tissue, indicate that the overall complex mechanical setting renders a rather uniform direction of cardiac fibers [9]. In turn, this allows strong conduction and optimal distribution of the contractile forces of the heart, which is further facilitated by the mechanical anisotropy of the tissue [3].

The development of a reliable cardiac model in vitro requires careful consideration of the mechanical behavior, function, and spatial architecture of cardiac cells in a dynamic microenvironment [10]. Indeed, under static and stress-free conditions, the mechanical microenvironment of cardiac cells cannot accurately recapitulate in vitro. Several studies have shown that long-term culture of cardiac cells in 2D without mechanical stimulation leads to a loss of phenotype, functionality, and dedifferentiation capability [11]. On the contrary, the investigation of cellular benefits of mechanical stimuli on cardiac cells in vitro in both two-dimensional (2D) and three-dimensional (3D) cultures allowed the clarification of the role of mechanical signaling on the cardiac cell function and phenotype [12]. Stimulation in 2D cell cultures is based on various approaches employing either the stiffness of the substrate material or the mechanical stretching of the substrate. Overall, the favorable outcomes of these studies are associated with improvement in cellular and sarcomere organization, gene and protein expression, and functional properties [13,14,15,16,17,18]. This indicates the promising positive effect of the minute adaptation of the mechanical work environment on cellular phenotype in vitro cardiac models [19].

Investigations of the contribution of mechanical stimuli during the organization and alignment of cells, in general, and of cardiomyocytes, in particular, yielded conflicting results. Indeed, external cyclic stretch has been reported to lead to alignment of cells along the stretch direction in some 2D culture experiments, but in the *a priori* unexpected perpendicular direction in others [20,21]. Similarly, engineered heart tissues (EHT) [22] are associated with massive and robust alignment along the tangential direction which corresponds to the main stress direction, due to cellular contraction. Yet, under conditions of externally cyclic stretch, cellular orientation perpendicular to the stretch direction could be documented in some 3D hydrogel culture systems [12]. These conflicting results may arise through identical cellular mechanisms yet incompletely controlled boundary conditions [23]. The mechanism at play is thought to be strain avoidance, implying alignment of the cells along directions where contraction by the cells or externally imposed stretch imposes the least deformation [23,24]. Strain avoidance, previously linked to the thermodynamics of stress fiber assembly [24], explains the extensive cell orientation with macroscopic magnitudes in the engineered heart tissues [22]. Indeed, in this case, only the circumferential direction cannot be contracted; strain avoidance also naturally explains cell orientation along the direction of greatest stiffness on anisotropic substrates and may also be involved in topographical effects on structured surfaces. In cyclic stretch experiments, an experimental complexity that needs to be considered is that by controlling the uniaxial cyclic motion, a non-contractable direction is also imposed. The final alignment outcome depends on whether or not self-condensation is possible in the other directions and on the magnitude of the applied stretch [23,24].

The complex interaction between long-term contraction boundary conditions and cyclic mechanical stimulation may be more than just an experimental nuisance. We hypothesize here that this interaction could represent the basis of a physiological mechanism capable of triggering alignments off-axis regarding the preferred mechanical stretch direction. This would not only allow in vitro off-axis alignment engineering but also a better understanding of helicoidal arrangements observed in vivo.

From a practical perspective, this report aims to investigate the combined effects of mechanical stimulation and geometrical constraints imposed on 3D cardiac cell cultures through surface topography. To achieve this, uniaxial mechanical stretching was applied to pre-aligned 3D cultures of cardiac cells in a decellularized extracellular matrix (dECM)-fibrin hydrogel. The possibility of producing oblique alignments between 0° and 90° was evaluated to set the basis for future development of helicoidal structures, similar to those of the native heart. Furthermore, our study also addressed the quantification of beating characteristics and gene expression, paramount for the evaluation of cardiac functionality. A crisp theoretical model for understanding the oblique alignment angles based on numerical simulation and nonlinear strain avoidance by the cells constitute the basic hypothesis of this work (Figure 1).

## 2. Materials and Methods

### 2.1. Mechanical Stretcher Device

We implemented a custom device designed to culture cells in a standard CO_2_ incubator while applying controllable mechanical stretching. The apparatus consists of an actuation part and a flexible PDMS cell culture chamber (Figure 2). The mechanical actuator consisting of a stepper motor (5 mm, 1.8°, 0.06 Nm, QSH2818-32-07-006, Trinamic) was used to apply mechanical cyclic stretching motion to the Polydimethylsiloxane (PDMS) chamber (schematic: Figure 2A; mounted: Figure 2D). The transparent PDMS chamber dedicated for cell cultivation in 3D constructs can feature optional microgrooves on the surface (Figure 2C). Our previous work has outlined the methods used in the creation of microgrooves [26].

Electrical signals were transmitted via a control system located outside the incubator. This system consisted of an electrical board (step motor controller, TMCM-1110 STEPROCKER, Trinamic) wired to the mechanical stimulator, controlled by a user-friendly interface written in MATLAB (Matrix Laboratory). The MATLAB program allowed adjustments of elongation and frequency. The PDMS chamber was cast in a polytetrafluoroethylene (PTFE) mold (Figure 2B) before undergoing 2 h curing at 80 °C. Silicon masters for molding microgrooves were fabricated using direct laser writing photolithography and deep silicon etching employing a standard Bosch process at a channel size × space of 350 × 350 μm. The PDMS’ inner surface was treated with 100 W power oxygen plasma for 60 s to achieve an increased PDMS hydrophilicity. The chamber featured a fixed and an opposing movable side. The chamber was subjected to sterilization in autoclave to ensure sterility and prevent potential contamination of the cell culture.

### 2.2. Elongation Measurement

The digital image correlation (DIC) technique was employed to accurately characterize the elongation applied by the custom-built stretcher to the PDMS chamber and, thus, the cells, in order to ensure repeatable results. This approach relied on digital image correlation (DIC), which enables the mapping of an image onto a field of displacements aiming to obtain strain fields within a region of interest (ROI) for a material sample undergoing deformation [27]. As shown in Figure 3, random pattern of paint dots generated by spraying the PDMS chamber’s surface will be imaged during the deformation. Thus, DIC will obtain a one-to-one correspondence between dye points, in the initial undeformed picture and current configurations, by taking small subsections of the reference image, referred to as subsets (see Appendix A), and determining their respective locations in the deformed configuration. The smallest parameters capable of generating noise-free, precise mapping were selected. To efficiently avoid nonlinearities and out-of-plane effects during the computation of strain field, we selected an ROI that was not in overly close proximity to the grip and the chamber’s wall. DIC analysis was carried out with a subset radius of 40, a spacing of 5, and a radius of 15 [27].

### 2.3. Cell Culture

Two groups of cells were considered for the experiments: (i) H9c2 in co-culture with NOR-10 (fibroblast) cells as the first model to set up the system, and (ii) neonatal cardiac cells from rats for functionality tests. The H9c2 cell line was obtained from the European Collection of Authenticated Cell Cultures (ECACC) (Lot# 17A028) and cultured in DMEM medium supplemented with 10% fetal bovine serum (FBS), 1% penicillin and streptomycin in 75 cm^2^ tissue culture flasks at 37 °C, and 5% CO_2_ in humidified atmosphere. NOR-10 (ECACC 90112701) cells were obtained from the European Collection of Authenticated Cell Cultures. The cells were cultured in DMEM medium supplemented with 10% fetal bovine serum, 1% penicillin and streptomycin in 75 cm^2^ tissue culture flasks at 37 °C, and 5% CO_2_ in an incubator. The neonatal rat cardiac cells were purchased from Lonza (R-CM-561) and cultured in RCBM basal medium supplemented with 7.5% horse serum, 7.5% FBS, and 0.1% GA. The culturing was conducted as suggested by the supplier.

### 2.4. Hydrogel Preparation

The dECM-fibrin hydrogel was prepared as previously described [28]. Briefly, gel precursor was generated by mixing dECM and fibrinogen (Sigma, F3879) at a final concentration of 5 mg/mL and 26.4 mg/mL, respectively. In total, 1 × 10^6^ cells/mL were added to the gel precursor before supplementing with Thrombin (Sigma, T1063, 250 U/mL) and calcium chloride to trigger gelation. An amount of 500 µL of the resulting cell-suspended gel solution was rapidly cast into the PDMS chambers. Gelation occurred within 2 min. The medium was exchanged every 2 days. The same cells without hydrogel were cultured on the PDMS substrates as 2D control.

### 2.5. Mechanical Stimulation

Samples were incubated for 48 h before mechanical stimulation to allow cell attachment and spread either within the hydrogel or onto the plasma-treated PDMS surface. The cells in the in vitro model were subjected to mechanical stimulation at 15% elongation and 1 Hz frequency, mimicking the physiological levels of mechanical stimulation experienced by cells in the heart. Samples were subjected to this regimen for 2 h per day, over 7 days. Intermittent mechanical stretching stimulation has been shown to lead to higher tissue regeneration and improve the expression of cardiac-related genes and proteins compared to unstimulated counterparts [29,30].

### 2.6. Beating Characteristics

The contractile events of primary cardiomyocytes in 3D cultures were evaluated under an optical microscope and quantified using a video/time-lapse recording. Mechanical stretching was applied to primary rat cardiomyocytes that were cast into the PDMS chamber at a density of 1 × 10^6^ cells/mL in the dECM-fibrin gel precursor. Quantification of the mechanical beating rate of the cells relied on the acquisition of moving images with a video camera connected to the microscope [31]. Temporal peak detection was based on a custom ImageJ plugin in Java used to evaluate local beating frequency and temporal phase shift from the heatmap videos of the samples with and without mechanical stimulation (the details of this characterization can be found at the following link: https://github.com/tbgitoo/calciumImaging (accessed on 4 December 2022)).

### 2.7. Cell Immunostaining

α-actinin and connexin-43-specific immunofluorescence staining of samples subjected to or in absence of mechanical stretching was performed after 7 days to quantify and compare the expression of both proteins. Samples were washed with phosphate buffered saline (PBS) (Gibco 2062235) and fixed with 3% paraformaldehyde (PFA) for 15 min at room temperature, then washed with PBS and permeabilization with Triton 0.3% in PBS for 15 min. Phalloidin-Atto 488 (1:50) staining was conducted for 45 min at 4 °C in order to reveal the actin filaments. An additional washing step with PBS and blocking with PBS-BSA 1% for 10 min at room temperature was included for α-actinin and connexin-43-specific staining. Incubation with primary antibodies suspended in PBS-Tween 0.1%-BSA 1% occurred overnight at 4 °C. Washing was performed with PBS-Tween 0.1%-BSA 1%. The secondary antibody diluted in PBS-Tween 0.1%-BSA 1% was added to the sample for 1 h at 37 °C. Finally, the cells were washed and stained with DAPI for 5 min. Upon DAPI removal and PBS washing steps, cells were visualized under a ZEISS LSM 700 inverted confocal microscope.

### 2.8. Mechanical Simulations

Two types of mechanical simulations were employed. In a first step, static mechanical behavior of the flexible PDMS chamber, with and without cells, was stimulated to validate the reliability of strain transmission to the cells in 2D or in 3D hydrogels. In a second step, we carried out simulations of strain distribution under active cell contraction, in addition to the applied stretch, to better understand cell orientation.

For the first part, targeting numerical evaluation and validation of application of imposed stretch, we carried out stationary analysis in the Structural Mechanics Module of COMSOL MultiPhysics^®^ 5.6, with the physics interface selected as solid mechanics. Three different chamber geometries, employing settings either without cells, 2D, or 3D cell cultures, leading to a total of nine different cases, were investigated (Table 1). The corresponding 3D models of the different chambers were developed on SolidWorks, according to the dimensions of the actual PDMS chambers (Table 2). The difference between 2D and 3D cell culture in the grooved membranes is presented in Table 1 with green (2D) and red (3D) cells. For these simulations, the following assumptions were used: The behavior of all materials is linear, homogeneous, and isotropic, as shown in Table 3. The PDMS as a polymer and the dECM fibrin hydrogel, indeed, have a wide elastic region up to 40% strain. In accordance with the maximal physiological strain found in the human body, the H9c2 cells were placed under 20% strain.In the case of 2D geometry, the cell layer was modelled by a thin film that also rendered some concentration constraints. The model consisted of a clamped boundary condition on one end (fully built-in, translation, and rotation degrees of freedom set to 0) and a displacement in the stretching direction on the other end of U2 = 8 mm (15% stretching). These boundary conditions suppressed all possible rigid body modes. The load definition allowed for symmetry around the plane Oyz. In addition, the adhesion between the cell layers and the PDMS membrane was assumed to be strong enough to avoid peeling.The entire solid domain was meshed by swept triangular elements, resulting in hexahedral elements. The element size was set to extra fine (45 µm–1 mm). Refined meshes along the grooves (maximum element size of 0.1 mm) ensured computational accuracy. The mesh convergence of the stimulation was guaranteed for all cases.The strain field obtained with COMSOL MultiPhysics^®^ 5.6 for the chamber without grooves and cells was then compared to DIC analysis results.

Regarding cellular orientation, we restricted ourselves to the four cases of interest including the presence of cells in 2D and 3D, and parallel or perpendicular orientation between applied stretch and groove direction. In order to study the effect of cellular volume forces and perform custom strain addition between static and cyclic components, we used the finite element simulation environment Netgen. Netgen requires explicit variational formulation, but inherently allows full programmatic control (Python) over the simulation and evaluation.

For the simulation design of cellular orientation, we note that modeling of cytoskeletal stress fiber architecture [24] has led to the suggestion that both static strain components controlled by setup details and the intended cyclic stretch need to be taken into account [23].

Here, we use a simple, but nonlinear, strain addition model for the cumulation of cyclic and static strain effects. Cyclic stretch impacts on cell orientation indicate a nonlinear effect; that is, a purely linear response would average out the extensional and compressive components of purely cyclic stretch. Our aim was to implement the potential capability of stress fibers to explore each spatial direction into a lumped model for cell orientation, which ultimately provides a global orientation angle in the thermodynamically most favorable direction [24]. In each spatial direction, the relative length change of a stress fiber was considered. Thus, the overall cell orientation is represented by the direction with least compression. The nonlinearity indicates that the cyclic stretch component has a net effect by orienting the cells away from the most compressive direction during the compressive phase.

We link the appearance of strain concentration in conditions that are, at first sight, free of deformation to the phenomenon of self-condensation [12,22]. Self-condensation refers to the densification and reorientation of hydrogel components, and, concomitantly, of cells, arising through contractile forces deployed by the cells [22]. At seeding, with no anticipated preferred local spatial direction, a volumetric contractile force was already applied homogeneously throughout the hydrogel (3D) or cell layer (2D). Interestingly, in the 3D setting, an empirical and partial detachment of the hydrogels upon extended culture arises at the lateral side walls of the grooves. Here, we used an elastic spring force at lateral walls of the grooves to account for this partial detachment. Full details of the boundary conditions and variational implementation are provided in Appendix A.

In order to assess the proper addition of cyclic stretch and self-condensation strain, we again considered representative stress fibers and all spatial directions and their elongations, to which net elongations associated with the compressive phase of cyclic stretch (e_cyclic_) and self-condensation (e_stretch_) were added by a polynomial addition law:e_tot_= ((e_cyclic_)^−n^ + (e_self_)^−n^)^−(1/n)^(1)
where n is a small positive number. Here, n = 4 was used. The addition of negative powers of the deformations results in higher weight to compression (0 < e < 1) than to elongation (e > 1), which is the essence of compressive strain avoidance. The reversion to stretch dimensions via the ^−(1/n)^ element ensures that if the elongation effects are spatially aligned, the magnitude of the resulting elongation scales proportionally to the individual contributions.

In the stress addition law defined by Equation (1), the optimal cell orientation corresponds to the spatial direction with least compressive e_tot_, i.e., the direction with maximal e_tot_ values.

### 2.9. Statistical Analysis

The data was compared using an unpaired *t*-test (two-tailed, equal variances) and two-way ANOVA test with multiple comparison, and one-way ANOVA test with multiple comparisons in the GraphPad software. Error bars represent the mean ± standard deviation (SD) of the measurements (* *p* < 0.05, ** *p* < 0.01, *** *p* < 0.001, and **** *p* < 0.0001).

## 3. Results

### 3.1. Device Operation

We first evaluated whether our custom-built stretching device could reliably apply the intended stretch to cell culture chambers and, ultimately, to the cells. For this, the displacement fields for a PDMS chamber undergoing cycles of stretching by digital image correlation (DIC) were evaluated. This technique yields a grid containing displacement and strain information with respect to the reference configuration (Figure 4), also referred to as Lagrangian displacements/strains. For a region of interest (ROI) of 15 mm length, and an intended strain of 5%, we found an elongation magnitude of 0.6 mm, as compared to the anticipated value of 0.75 mm. The minor difference is most likely attributable to the clamping system used for joining the PDMS chamber to the stretching device.

Furthermore, the magnitude and distribution of the displacement field corresponded approximately to the values simulated in COMSOL MultiPhysics 5.6, used for the validation of the device operation principle.

Having resolved the DIC measurement-based validation of the basic device operation principle, we sought to determine the strain transmission to cells. Indeed, the DIC measurement and simulations of unstructured membranes suggest that, despite some losses due to the clamping system, the applied strain is reliably transmitted to the main membrane area as desired. However, the presence of the grooves, cells, and, optionally, the hydrogel, in the case of 3D geometries, requires further investigation. In order to study strain transmission to the cells, we considered the nine possible combinations of cell seeding approach and topography (Table 1). These combinations arose as a result of a combination of three different cell seeding approaches (i.e., unseeded vs. seeded in 2D or 3D), and three topographies (unstructured, grooves parallel and perpendicular to stretch). For each geometry, we numerically evaluated the stretch resulting from the application of an intended 36% extension at a representative location (near chamber center, on the groove midline and, in 3D, at groove half-height).

Table 1 demonstrates that the stretch is reliably transmitted to cells at such representative locations, albeit with an enhancement factor of about 2× due to the stress concentration in the particular geometry of perpendicular grooves. Most importantly, in all cases taken into study, the main stretch direction and order of magnitude perceived by the cells correspond to the intended stretch.

### 3.2. Cell Alignment

Having validated the device operation and transmission of the intended stretch to the cells, we investigated the alignment of cells in 2D or 3D in vitro cultures as a function of topography and of mechanical stimulation. To mimic the mixed stromal and cardiomyocyte composition in the native heart, H9c2 cardiac cells were co-cultured with Nor-10 human fibroblasts in unstructured 3D dECM-fibrin hydrogels. The employment of Nor-10 fibroblasts permitted not only observation of differentiation but also the homogeneous distribution in the absence of any specific orientation [28]. The placement of hydrogel into microgrooves with a square cross section of 350 × 350 µm resulted in a robust cell orientation along the groove direction and throughout the entire microgroove volume. Similarly, cells seeded in 2D onto the grooved structure aligned with the microtopography. At the same time, no specific orientation could be documented in the absence of microstructuring.

Here, the mechanical stretch was superimposed onto this topographical alignment. Interestingly, the 2D cultivation setting featuring unstructured PDMS substrates is associated with cyclic mechanical stretch (15%, 1 Hz) that strongly orients the cells perpendicularly to the stretch direction on flat PDMS substrates (Figure 5H), in accordance with the principle of strain avoidance [21,32]. Upon culturing cells on patterned substrates, the stretching dominates and, furthermore, a cell orientation nearly perpendicular to the axis of stretch was observed (Figure 5A–C). This perpendicular orientation is, indeed, observed regardless of parallel or perpendicular orientation of the grooves toward the direction of stretch, establishing, in this case, the dominance of the effect of stretching.

Next, we examined the influence of mechanical stretch on 3D co-cultures of H9c2 and Nor-10 fibroblasts. Mechanical stimulation of cells in hydrogel without patterning showed partial perpendicular orientation to the stretch axis (Figure 5D, *p* = 0.013 against 45° for random orientation between 0° and 90°). Thus, the alignment direction perpendicular to stretch observed in 2D also applies to 3D cultures. Quantitatively, the alignment effect of stretching in the absence of patterning is weaker in 3D than in 2D cultures. We ascribe this difference between the 2D and 3D geometry to partial inefficiency in coupling mechanical stimulation into hydrogels with unstructured substrates, where partial slippage between hydrogel and PDMS substrate is more likely.

We then proceeded to mechanical stimulation of cells cultured in the 3D hydrogels featuring 350 × 350 µm cross-section grooves. This scenario intentionally combines mechanical stretch and topographical alignment cues offered by the grooves. The results reveal an intriguing interplay between the two cues. Application of cyclic stretch in the same direction of the microgrooves rendered the orientation of cells at an angle of around 40° to 60° to the direction of mechanical stretching (Figure 5F). On the contrary, if cyclic stretch was applied perpendicularly to the grooves, the orientation of cells along the grooves matched the orientation imposed by the stretch and, thus, an alignment perpendicular to stretch was observed (Figure 5E). Overall, our observations on cell orientation indicate that strain avoidance involves a cellular integration mechanism for independent geometric and mechanical stimuli, producing oblique angles, as outlined in Figure 5G.

### 3.3. Beating Behavior of Cells under Mechanical Stimulation

In order to examine whether mechanical stimulation was compatible with normal function of cardiomyocytes, rat primary neonatal cardiomyocytes were cultured in our dECM-fibrin hydrogel system. Time-lapse recordings of neonatal cardiac cells cultured in the presence or absence of mechanical stimulation for 7 days enabled the evaluation of local frequency and local phase (Figure 6C,D).The results indicate well-synchronized cultures in both conditions. Importantly, a higher local frequency was observed in the samples subjected to mechanical stimulation. Furthermore, the beating rate documented after 10 days of culture of mechanically stimulated samples was 2 times higher than in the non-stimulated counterparts (120 bpm vs. 60 bpm).

### 3.4. Cardiac Markers Expression

Supporting improved beating function, enhanced expression of typical cardiac marker proteins has also been investigated under mechanical stimulation (Figure 5B and Figure 6A). Therefore, we analyzed the expression of α-actinin as a marker of the contractile apparatus and connexin-43, a marker for cellular interconnectivity. Protein expression levels’ percentages were quantified and represented as surface area covered by cells expressing the marker relative to the total area of the fluorescent image. This approach assumed minimal cell growth at near 100% confluency. Our investigations revealed substantially elevated expression of α-actinin and connexin-43 (*p* = 0.0017 for α-actinin and 0.00013 for connexin-43) in samples subjected to mechanical stimulation. Hence, exogenous mechanical stimulation boosts cardiomyocyte maturation in 3D cultures, in agreement with the results of others [12].

### 3.5. Additive Strain Avoidance Model

Having confirmed that cyclic mechanical stimulation not only modulates cellular alignment but has also favorable biological effects, we sought to characterize the local mechanics behind the observed cellular alignments under various conditions.

Here, an additional simulation in Netgen/NGSolve addressing cellular contractility was employed. Seeding is expected to lead to homogeneous volumetric contraction forces, in the case of 3D cultures and, essentially, isometric biaxial contraction for cells seeded as a 2D layer on PDMS substrates. Depending on the local mechanical surrounding, cellular contraction is expected to lead to asymmetric local deformation. The principles of strain avoidance then dictate the preferred cellular alignment along the direction that is least compressed. Here, we mainly focused on the mechanical environment within the grooves. Our investigations show that, while generally following the grooves, the hydrogel undergoes some contraction with regional/local detachment from the lateral sidewalls in a 3D setting. This could be modelled by employing a flexible spring force that allows some flexibility for lateral contraction across the grooves. Thus, the microgrooves are modelled, in part, in a similar manner as supported cell-condensing structures, such as the ones used to align cells in EHT. The least compressive strain direction, in this case, is mostly along the grooves, and leads to longitudinal cell alignment (Figure 7B). Given its relevant compressive half-cycle, the cycle stretch imposes a cellular orientation perpendicular to the main stretch direction (Figure 7A, with global alignment perpendicular to the longitudinal cyclic stretch).

The achieved preferred orientation is a result of the nonlinear addition of the cyclic and static components, which was further carried out based on nonlinear sensitivity to elongation, according to eq. 1. The approach was implemented/applied to the four conditions involving grooves and mechanical stretch, as presented in Table 1, i.e., combinations of 2D and 3D seeding on grooved substrates with mechanical stretch either along the groove direction or perpendicularly. Figure 7C shows the nonlinear strain addition principle based on the combined deformation of a unit circle probe with radial length change addition via Equation (1) with n = 4, for the 3D configuration with stretch along the grooves, giving rise to oblique cell alignment (Figure 7D) in the grooves. Table 4 summarizes the expected and measured orientation for all four relevant configurations. From a quantitative perspective, the simulations are in line with the observed orientation directions. Altogether, the softness of hydrogel in the 3D setting indicates that cellular contraction and externally applied stretch result in local strains with a similar order of magnitude. Furthermore, when the orientation imposed by the stretch and self-condensation differ, an intermediate oblique angle can be achieved. This particularly occurs at the site where cells are less exposed to compressive strain rather than along either one of the main directions. Given that PDMS substrate is substantially stiffer than the cells in 2D configurations, cells can impose only main deformation. The stretching device is not limited by the stiffness of PDMS and, therefore, imposes similar large-scale deformations as on the hydrogel. As a result, the simulations indicate that externally applied stretch featuring large amplitude should almost completely dominate the behavior of cells in the 2D scenario, in agreement with the observed results.

## 4. Discussion

The native physiological heart muscle twists along its long axis because of the opposite rotation of the subepicardium and subendocardium. Structurally, this is reflected by a helical arrangement of cell and fiber alignment throughout the ventricle walls. Hence, reproducing and stabilizing this helical arrangement would impact the functionality of the cardiac model in vitro and, furthermore, renders a fundamental interest in engineering biomimetic models and implants. To achieve a gradually oblique alignment of cardiac cells in 3D native heart tissue, we focused on the impact of simultaneous application of mechanical stimulation and hydrogel patterning on cardiac cell functionality, orientation, and protein expression.

Our results demonstrate that alignment at oblique angles with respect to the applied stretch can be robustly produced upon application of cyclic mechanical stimulation along the direction of microgrooves filled with cell-laden hydrogels. Indeed, when cyclic stretch and grooves are in the same direction, the applied mechanical strain tends to reorient the cells perpendicularly to the direction of stretching. At the same time, microgrooves promote cell alignment in the direction of their axis. As a result of these two constraints, the cardiac cells displayed an oblique orientation (around 45°) in 3D-patterned hydrogel in which the mechanical stimulation was applied in the direction of the microgrooves. These results indicate an additive interaction between the alignment mechanisms and are in line with previous reports combining conflicting alignment cues in 2D [33]. In this study, which used a myoblast cell line, an intermediate angle at 47.9°, reflecting competition of chemical patterning and cyclic stretch, was found. Upon application of mechanical stretch perpendicular to the groove direction, the two alignment cues were found to promote similar alignment along the grooves, while no oblique alignment could be documented. In our 2D control experiments, the mechanical stretch nearly completely dominated the cellular orientation.

The combination of mechanical stimulation and surface topography in 3D improves the recapitulation of the in vivo conditions, as judged by the enhanced expression of the maturity markers α-actinin and connexin-43. This means that the stimulation improves the integrity and maturation of the 3D constructs. Moreover, the beating analysis of 3D structures in the absence or presence of mechanical stimulation confirms the higher beating rate and improved integrity of cardiac cells in dynamic conditions. Hence, a dynamic 3D environment provides cardiac cells with improved conditions that promote maturity and functionality, and, furthermore, provide a more physiologically relevant cardiac model in terms of orientation. This is in line with previous reports suggesting favorable biological effects of mechanical stimulation in cardiomyocyte culture [12]. In addition, it rules out that oblique alignment would render an ill effect onto mechanical stimulation.

Detailed simulation of mechanical deformation generated by cellular contraction and mechanical stretch sheds light on local mechanical forces and deformation, and, thus, on possible cellular integration mechanisms behind the macroscopically observed additivity of the conflicting alignment cues. Based on the notion that under suitable dynamic conditions, cellular stress fibers depolymerize under compression, but elongate under extension [24], the concept of strain avoidance has been coined [23]. Strain avoidance tends to provide a cue towards alignment perpendicular to the mechanical stretch direction. Upon 2D cultivation of cells on top of the PDMS chips, the perpendicular alignment cues completely dominate cellular behavior. Our simulations suggest the ability of cells to deform the stiff PDMS to a small extent only, while the chip essentially transmits the entire applied strain to the cells. In 3D, cells are embedded in a soft hydrogel, on the strains produced by cellular forces which are competitive with the applied external strain. In conflicting situations, large oblique alignment angles can be obtained. According to our lumped cellular strain avoidance model, these would correspond to the least directions of least effective compressive strain.

In native cardiac tissue, there is a gradual, rather than abrupt, change of orientation. Therefore, our large off-axis orientations are, to some extent, unphysiological, hence the aim to provide a proof-of-concept experiment with large observable effects. Likewise, the strain avoidance is not the only cellular alignment mechanism at play. It has been previously reported that nanotopology [34,35], chemical [2] and stiffness gradients, and direct force and signal transmission by neighboring cells [12] also play important roles. Nevertheless, a mechanism capable of creating off-axis alignment is fundamental for the generation of helical structures. Hence, both our empirical and mathematical model can be considered as a first-of-its-kind in vitro model to willfully control oblique cellular alignment in 3D. Our model may also recapitulate and, thus, permit the investigation of an unexpected possible relation between the dynamic mechanical niche of cardiomyocytes and helicoidal organization in vivo.

## 5. Conclusions

Overall, our results of 3D cell alignment, both with mechanical stimulation and groove constraint, reveal a novel potential mechanism for the generation and optimization of helicoidal structures in the myocardium. Based on the insights outlined here, it is possible to experimentally achieve intermediate angles to a preferred direction. While cyclic stretch produces a perpendicular orientation, geometric alignment in grooves generates, instead, a parallel one. Nonlinear cellular addition of the strains, along with the strain avoidance concept, successfully predicts preferred cellular alignment under our 2D and 3D culture conditions. In our experimental model, cyclic stretch is fully compatible with cell survival and development in 3D dECM-fibrin. Maturation markers are, indeed, enhanced in neonatal cardiomyocytes under mechanical stretching conditions, and the beating culture period is extended. This demonstrates improvement in the functionality of the model by applying mechanical stimulation to a pre-aligned cell structure. In conclusion, this work reveals a possible rational basis for understanding and engineering oblique cellular alignment, such as the helicoidal one observed in the heart, using approaches that simultaneously enhance cardiomyocyte maturation and function.

## Figures and Tables

**Figure 1 bioengineering-10-00266-f001:**
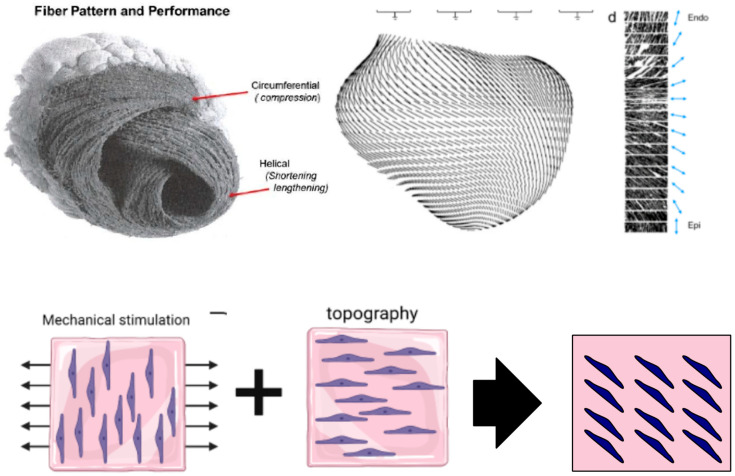
Oblique cell alignment. In the native heart, cellular arrangement is strikingly helicoidal [1,25], with orientation angle changing throughout the heart muscle wall, from endocardium to epicardium. In this work, we address the question of whether combined geometrical and mechanical stimulation can produce cell alignment at oblique angles to the main stress directions, in order to reproduce in vitro oblique cell alignment geometries reminiscent of the organization in the heart.

**Figure 2 bioengineering-10-00266-f002:**
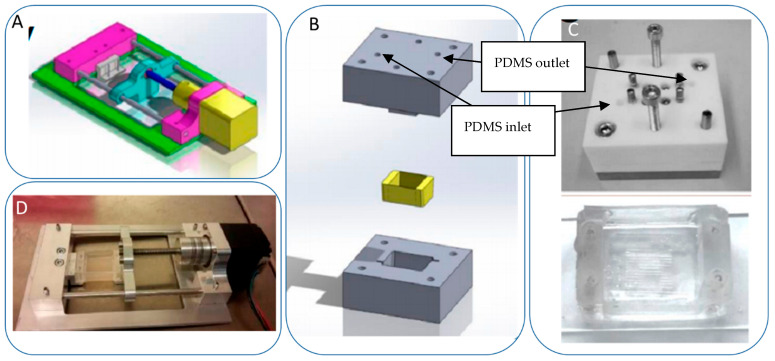
Schematic of the mechanical stretching device. (**A**) Schematic of the mechanical system; the yellow part shows the step motor, the light blue part is the movable side, the purple part is the fixed side, and the dark blue part is the screw which connects the movable side to the step motor. (**B**) Schematic of the PTFE mold for PDMS (in gray) and PDMS chamber (in yellow). (**C**) Realization of mold and chamber, including grooves at the bottom of the flexible PDMS chamber; PDMS is injected to the mold from the inlet hole using a 5 mL syringe. After filling the mold, PDMS is cured by placing it at 80 °C for 2 h. (**D**) Assembled device and PDMS chamber, fixed between two sides.

**Figure 3 bioengineering-10-00266-f003:**
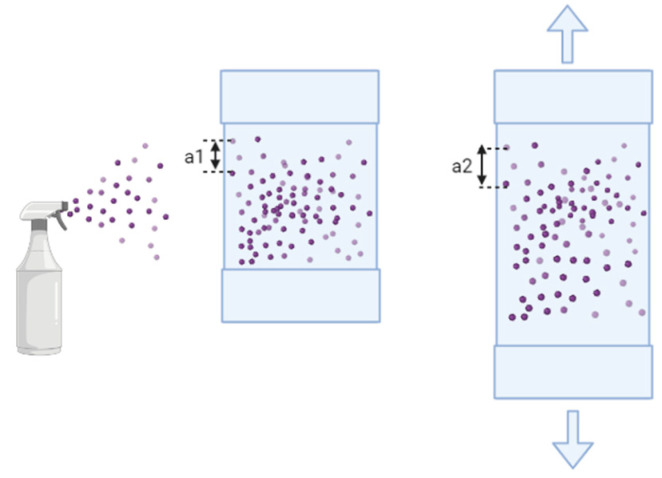
Schematic of preparation of PDMS chamber for DIC analysis. The dye is sprayed on the surface of the PDMS chamber. The displacement of selected dye dots was measured before and after the mechanical stretching.

**Figure 4 bioengineering-10-00266-f004:**
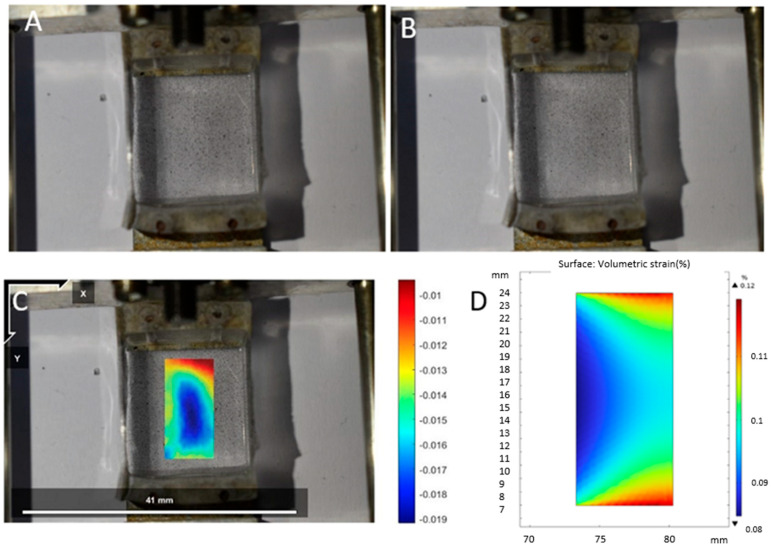
Evaluation of geometric deformation by digital image correlation (DIC). (**A**) Unstrained (native state) and (**B**) 5% strained state of PDMS chamber including DIC markers (paint dots). (**C**) Strain evaluated in the DIC experiment. (**D**) Strain simulated in COMSOL. For the DIC experiment, the images were obtained with a Nikon D5600; thanks to the intervalometer mode, one photograph is captured per second during several stretch cycles. The strain is set to 5%, velocity is set to 100, and the number of cycles is 5.

**Figure 5 bioengineering-10-00266-f005:**
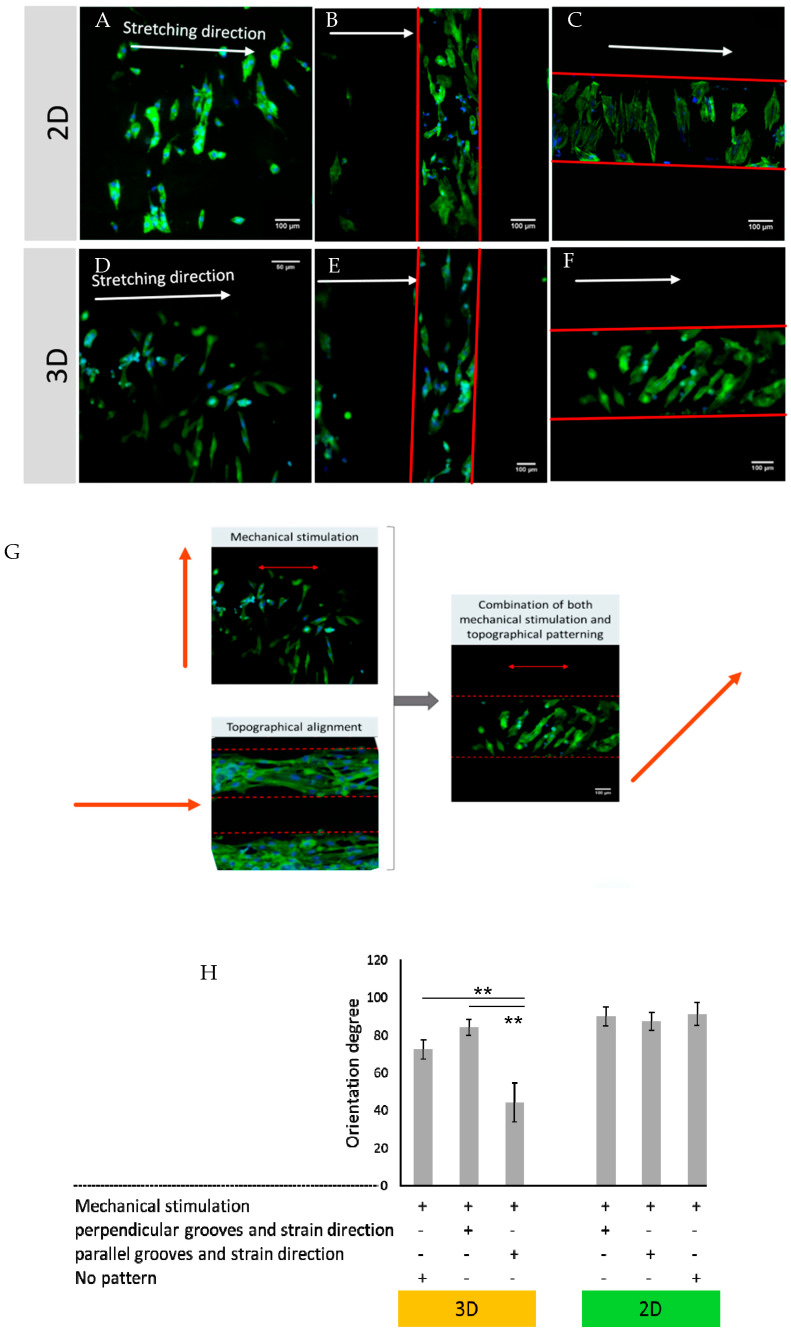
H9c2 co-cultured cells in 350 μm width and height grooves, stained with Phalloidin and DAPI, with cell concentration of 1 million cells/mL, stimulated under: (**A**) 2D culture without pattern; (B) 2D culture with pattern perpendicular to the direction of stimulation; (C) 2D culture with the pattern parallel to the stretching; (**D**) 3D condition without pattern; (**E**) 3D condition with pattern perpendicular to the direction of stimulation; and (**F**) 3D condition with the pattern parallel to the stretching (15%, 1 Hz). (**G**) The competition between mechanical and topological patterning. (**H**) The degree of orientation in different culture conditions (orientation degree is related to the angle between cell orientation and stretching direction). Error bars represent the mean ± standard deviation (SD) of the measurements (** *p* < 0.01).

**Figure 6 bioengineering-10-00266-f006:**
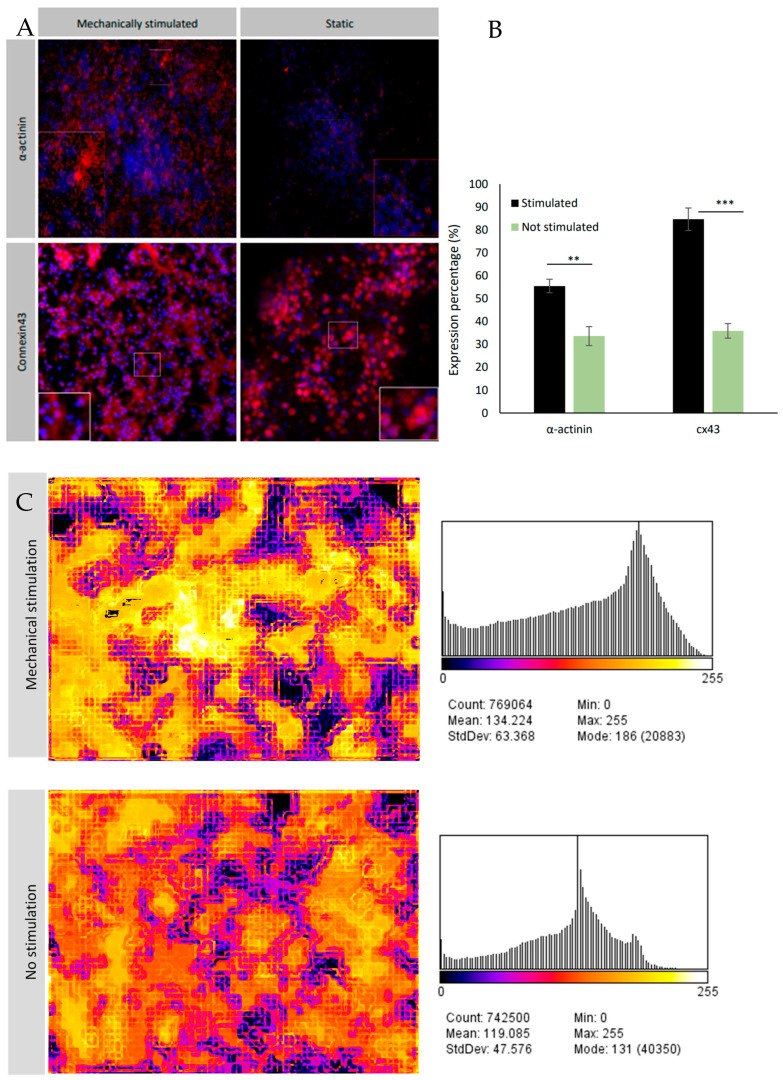
The expression of α-actinin and connexin-43 in neonatal cardiac cells cultured in 3D dECM-fibrin hydrogel under mechanical stimulation and static condition (without mechanical stimulation). (**A**) The immunofluorescence staining of the neonatal cardiac cells, (**B**) the area of the expression (red area) to the whole area of the image. This ratio was considered as the expression percentage. (**C**) Local frequency of neonatal cardiac cells cultured in 3D hydrogel, with and without mechanical stimulation (left). An unpaired *t*-test analysis (two-tailed, equal variances) of these data shows no significant difference between the two groups. (**D**) Local phase distribution with and without mechanical stimulation (right). An unpaired *t*-test analysis (two-tailed, equal variances) of these data shows no significant difference between the two groups. Error bars represent the mean ± standard deviation (SD) of the measurements (** *p* < 0.01, *** *p* < 0.001).

**Figure 7 bioengineering-10-00266-f007:**
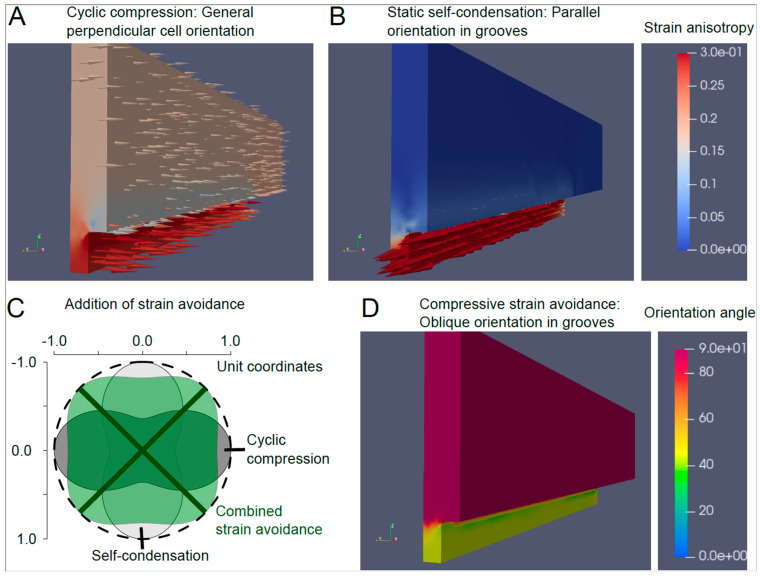
Simulation of the cell orientation: (**A**) in response to the cyclic compression, the cells orient perpendicular to the force direction. (**B**) Due to static self-condensation, the cells orient parallel to the grooves orientation. (**C**) The simultaneous application of the cyclic stretch and grooves’ pattern provides a combined strain avoidance which orients the cells to 45 degrees from the compression direction. (**D**) Oblique orientation in the grooves is the result of combined stain avoidance.

**Table 1 bioengineering-10-00266-t001:** Presentation of the 9 different geometrical and cell seeding cases that were analyzed: without grooves, with grooves parallel to stretching, with grooves perpendicular to stretching, with no cells, with 2D cell layer, and with cells 3D seeded in a dECM fibrin hydrogel (applied stretch in all cases is 36%).

	No Cell	2D Cell Seeding 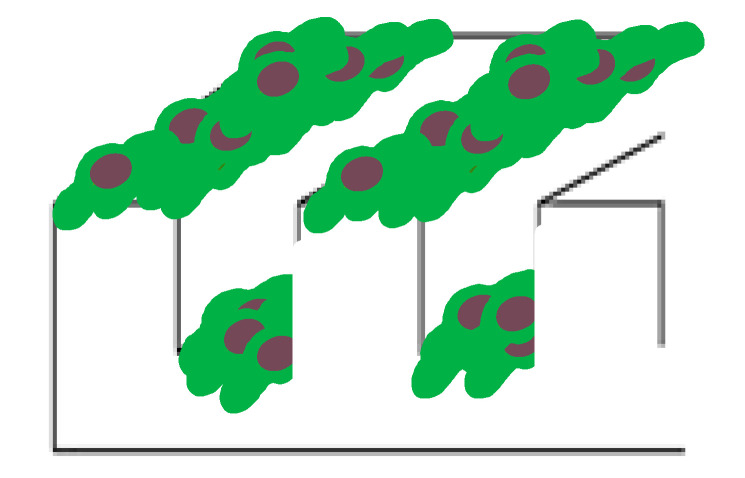	3D Cell Seeding 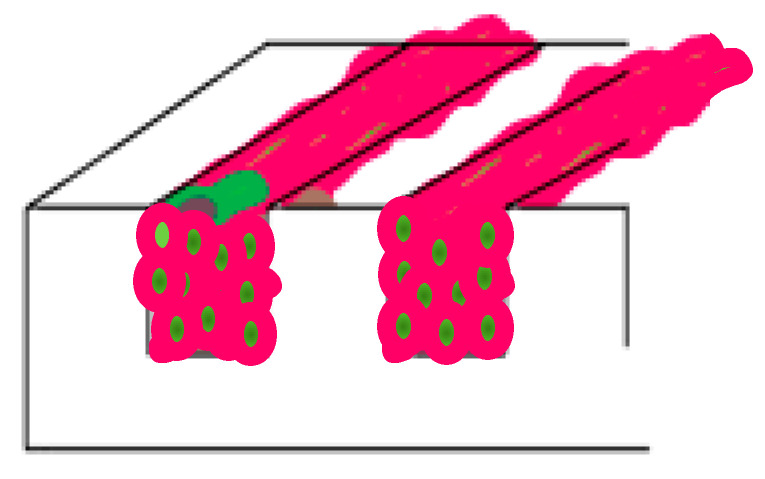
No groove Measured strain	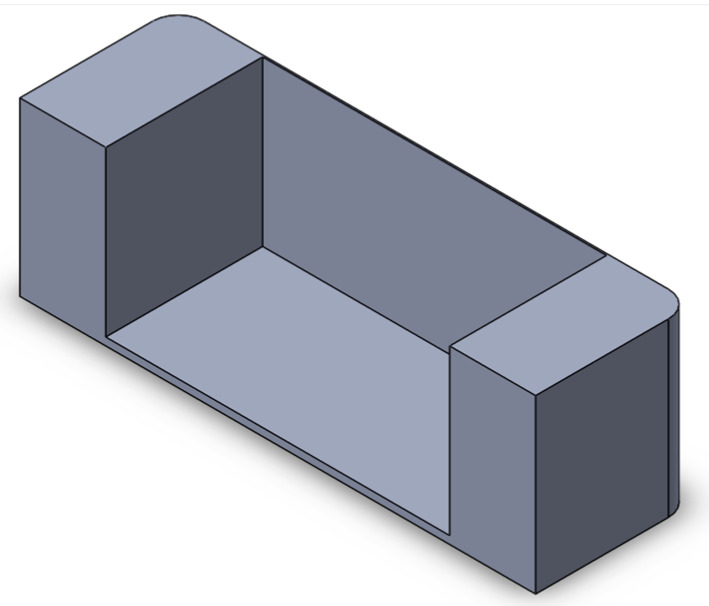	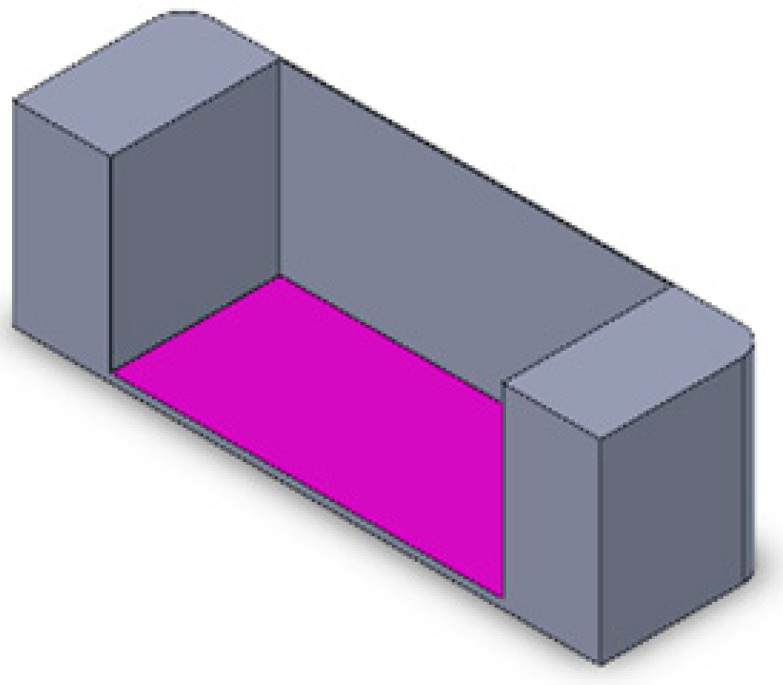 30.7%	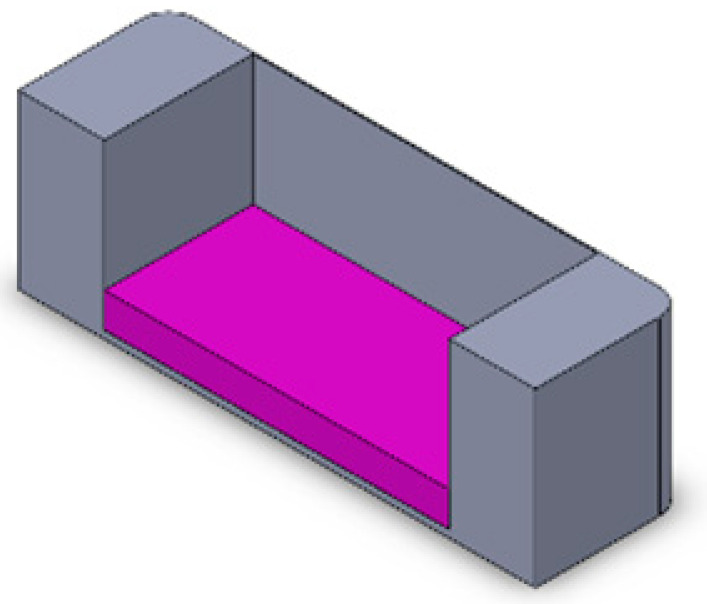 43.8%
Parallel grooves Measured strain	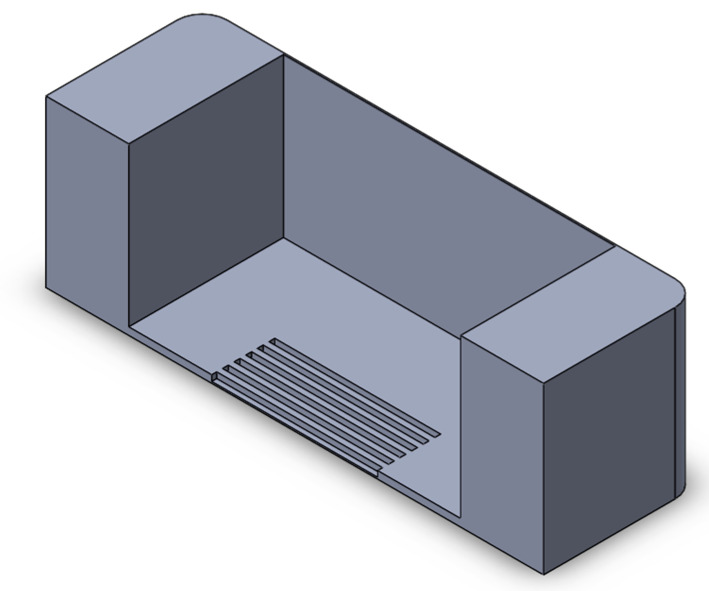	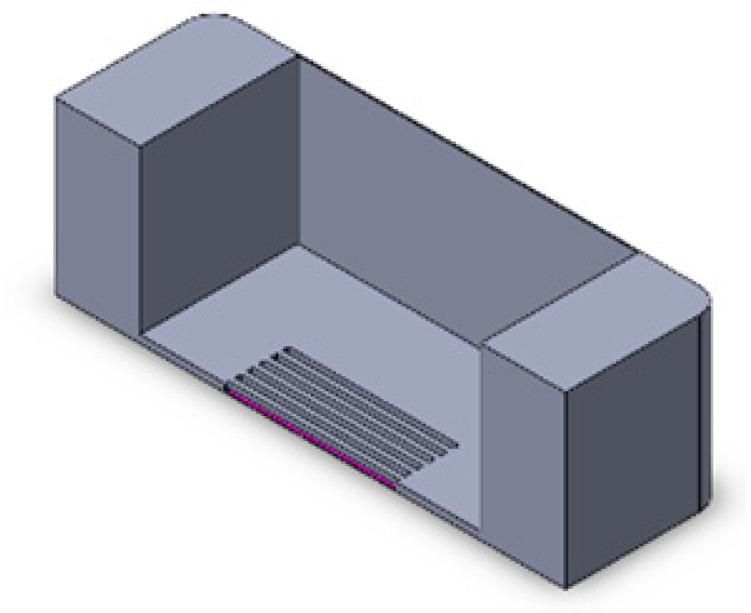 43.8%	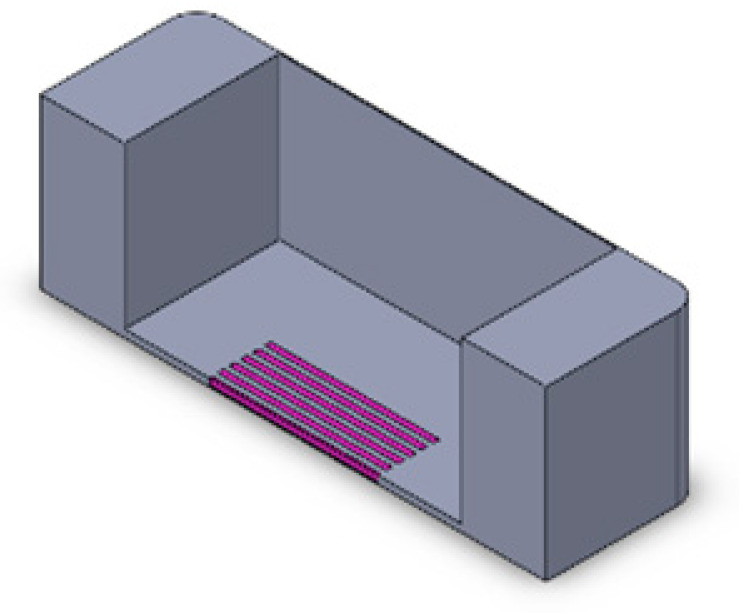 47.3%
Perpendicular grooves Measured strain	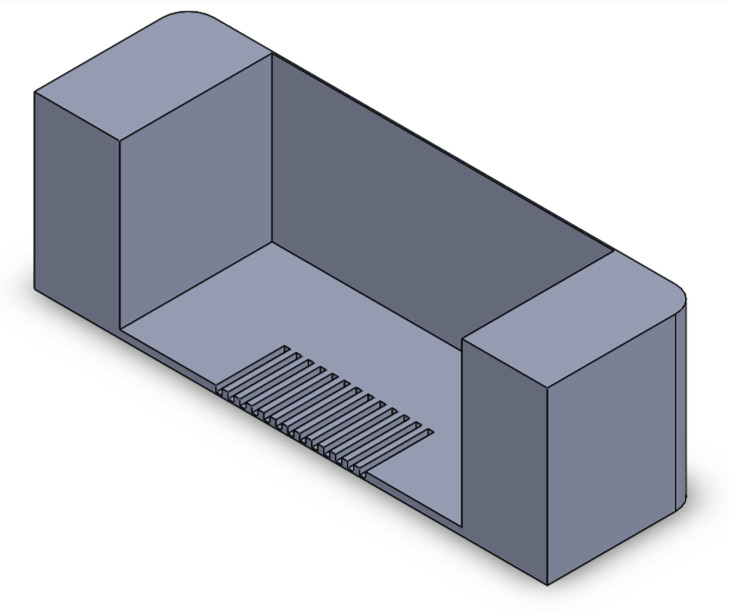	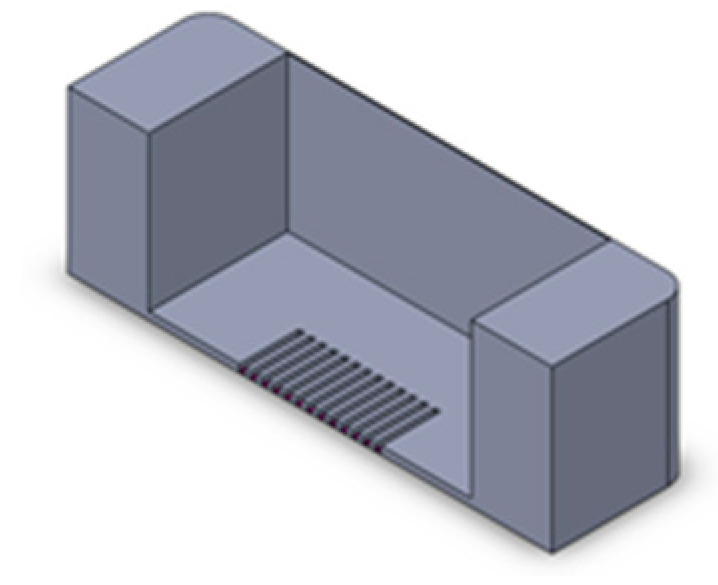 72%	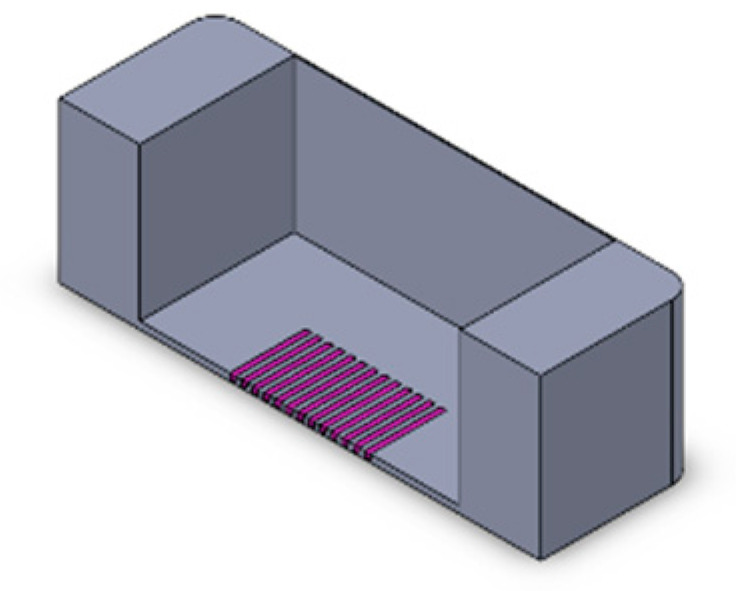 121%

**Table 2 bioengineering-10-00266-t002:** PDMS chamber dimensions.

Chamber length	20 mm
Chamber width	10 mm
Chamber height	10 mm
Thin sidewall thickness	0.1 mm
Thick sidewall thickness	5 mm
Bottom wall thickness	2 mm
Grooves dimensions	
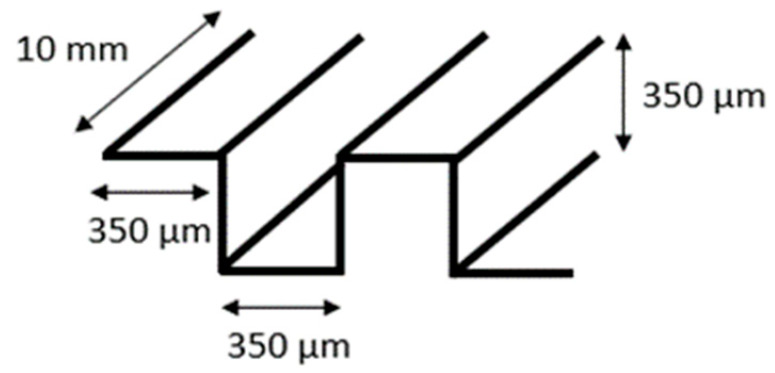	

**Table 3 bioengineering-10-00266-t003:** Materials’ mechanical properties.

	Young’s Modulus	Poisson’s Ratio	Density
H9C2 cells	30 kPa	0.5 (0.49 for computation)	1060 kg/m^3^
dECM-fibrin hydrogel [28]	21 kPa	0.2	
PDMS (COMSOL MultiPhysics^®^ 5.6)	0.98 MPa	0.5 (0.49 for computation)	965 kg/m^3^

**Table 4 bioengineering-10-00266-t004:** Theoretical and measured alignment angles. The angles here are expressed in degrees and indicate the orientation of the cells relative to the applied cyclic mechanical stretch (0° = aligned with stretch, 90° = perpendicular to stretch).

	Geometry: 350 Micrometer Grooves	
Cell Seeding	Stretch along Grooves	Stretch Perpendicular to Grooves
2D	Modelled: 84°; observed: 89.14°	Modelled: 89°; observed: 91.42°
3D	Modelled: 43°; observed: 45.71°	Modelled: 90°; observed: 86.85°

## Data Availability

Not applicable.

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
