# Peer review of "Toward a Physiologically Relevant 3D Helicoidal-Oriented Cardiac Model: Simultaneous Application of Mechanical Stimulation and Surface Topography"

_bioengineering, 2023, doi:10.3390/bioengineering10020266_

Round 1
Reviewer 1 Report
The paper entitled „Toward a physiologically relevant 3D helicoidal oriented car- diac model: simultaneous application of mechanical stimula- tion and surface topography” is an outstanding paper.
The method of the study fits well to the intended purpose. It is interesting that two simultaneous tendency was expected: applied cyclic mechanical strain would tend to reorient the cardiac cells perpendicularly to the direction of stretching while microgrooves would promote the cell orientation in the direction of their axis. However, in the experiments alike alignment along the grooves were experienced. It is also an important experience that mechanical stimulation improved the integrity and maturation of the 3D model.
The paper is recommended to accept as it is.
Author Response
We very much thank the reviewer for appreciating our study as “outstanding”. We keep thanking the reviewer also for finding our methodology and strategy well suited to the scope of the study and furthermore, to regard the findings as interesting. It is always encouraging to see that our efforts have made a positive impact and that our findings have generated further interest in the field.
Reviewer 2 Report
The manuscript is well-written and the methods are described in detail. Some minor corrections are
Please check the font and font size in the manuscript- all tables, figures, and table and figure legends are in a different font. Even the text has different
The citations are not as per MDPI format.
Where the fibroblasts were isolated from.
Author Response
Dear reviewer,
Thank you very much for your comments. Here, you can find our answers.
The manuscript is well-written and the methods are described in detail. Some minor corrections are
Thank you very much for considering our manuscript as “well-written” and providing the sufficient detail in order to benefit the reader.
Please check the font and font size in the manuscript- all tables, figures, and table and figure legends are in a different font. Even the text has different
We are thankful for your comment and apologize for this lack of attention to detail. The necessary modifications have been made in the revised version of the manuscript.
The citations are not as per MDPI format.
We greatly appreciate your feedback and have thoroughly reviewed the citation format in the manuscript, and addressed accordingly.
Where the fibroblasts were isolated from.
Thank you very much for this question. The fibroblasts were obtained from the European Collection of Authenticated Cell Cultures. These cells are isolated from skeletal muscle. The specifics of these cells are added in section 2.3. “NOR-10 (ECACC 90112701) cells were obtained from the European Collection of Authenticated Cell Cultures. The cells were cultured in DMEM medium supplemented with 10% fetal bovine serum and 1% penicillin and streptomycin in 75 cm2 tissue culture flasks at 37°C and 5% CO2 in an incubator.”
Best regards,
Authors
Reviewer 3 Report
Reviewing the manuscript entitled, “Toward a physiologically relevant 3D helicoidal oriented cardiac model: simultaneous application of mechanical stimulation and surface topography” by Navaee F er al., this is an article focusing on the investigation of the combined effects of mechanical stimulation and geometrical constraints imposed on 3D cardiac cell cultures through surface topography in in vitro experiments. It has potential as an experimental model for the future treatment of heart failure and elucidation of the pathophysiology of cardiomyopathy. Therefore, the authors need to respond to the following concerns.
The authors should correct the structure of the text. It is difficult to distinguish between the text and the figure legend.
Figure 2 requires detailed photos and legends.
What is MATLAB at line 146. The authors need to add an abbreviation table.
In 2.2. Elongation measurement, for better clarity, the authors should put illustrations or figures.
In 2.3 Cell culture, why did the authors use H9C2 cells and rat cardiomyocytes in separate experiments? H9C2 cell is a cell line established from the rat heart and is a cell line with properties of cardiomyocytes and skeletal muscle cells. When the author uses this, it is necessary to confirm whether your H9C2 cell has cardiomyocyte properties.
Human cardiomyocytes are also available for purchase, but the reason for using rat myocardium or H9C2 cells must be stated.
In 2.5. Mechanical stimulation, the authors mentioned “the native heart”. This is the in vitro experiments. You never used the native heart, you used cell line or primary cultured cell. The authors need to modify it.
The authors should state the rationale for cell stimulation for 2 hours per day for 7 days.
The authors need to describe the statistical analysis method. In Figure 5C, is there statistically different? The authors need to describe it.
It is difficult to understand whether this manuscript should be a technology-oriented manuscript or whether the obtained results should be documented. If it is the former, each section of 2. Materials and Methods needs detailed illustrations or figures. If this should document the obtained results, the preliminary sentences that seem to determine the experimental conditions shown at the beginning of the results section should be omitted.
Author Response
Dear Reviewer,
We really appreciate your feedback. Please see the attachment for our answers to your comments.
Best regards,
Authors

Round 2
Reviewer 3 Report
This reaches to an acceptable quality. Congrats!!